# Encapsulation of Bioactive Compounds for Food and Agricultural Applications

**DOI:** 10.3390/polym14194194

**Published:** 2022-10-06

**Authors:** Giovani Leone Zabot, Fabiele Schaefer Rodrigues, Lissara Polano Ody, Marcus Vinícius Tres, Esteban Herrera, Heidy Palacin, Javier S. Córdova-Ramos, Ivan Best, Luis Olivera-Montenegro

**Affiliations:** 1Laboratory of Agroindustrial Processes Engineering (LAPE), Federal University of Santa Maria (UFSM), Cachoeira do Sul 6508–010, Brazil; 2Grupo de Investigación en Bioprocesos y Conversión de la Biomasa, Universidad San Ignacio de Loyola, Lima 15024, Peru; 3Grupo de Ciencia, Tecnología e Innovación en Alimentos, Universidad San Ignacio de Loyola, Lima 15024, Peru

**Keywords:** polymers, encapsulation efficiency, gum Arabic, chitosan, coating material, bioactive compounds

## Abstract

This review presents an updated scenario of findings and evolutions of encapsulation of bioactive compounds for food and agricultural applications. Many polymers have been reported as encapsulated agents, such as sodium alginate, gum Arabic, chitosan, cellulose and carboxymethylcellulose, pectin, Shellac, xanthan gum, zein, pullulan, maltodextrin, whey protein, galactomannan, modified starch, polycaprolactone, and sodium caseinate. The main encapsulation methods investigated in the study include both physical and chemical ones, such as freeze-drying, spray-drying, extrusion, coacervation, complexation, and supercritical anti-solvent drying. Consequently, in the food area, bioactive peptides, vitamins, essential oils, caffeine, plant extracts, fatty acids, flavonoids, carotenoids, and terpenes are the main compounds encapsulated. In the agricultural area, essential oils, lipids, phytotoxins, medicines, vaccines, hemoglobin, and microbial metabolites are the main compounds encapsulated. Most scientific investigations have one or more objectives, such as to improve the stability of formulated systems, increase the release time, retain and protect active properties, reduce lipid oxidation, maintain organoleptic properties, and present bioactivities even in extreme thermal, radiation, and pH conditions. Considering the increasing worldwide interest for biomolecules in modern and sustainable agriculture, encapsulation can be efficient for the formulation of biofungicides, biopesticides, bioherbicides, and biofertilizers. With this review, it is inferred that the current scenario indicates evolutions in the production methods by increasing the scales and the techno-economic feasibilities. The Technology Readiness Level (TRL) for most of the encapsulation methods is going beyond TRL 6, in which the knowledge gathered allows for having a functional prototype or a representative model of the encapsulation technologies presented in this review.

## 1. Introduction

The characteristics of foods depend on several factors, such as forms of presentation, nature, texture, flavor profiles, and major composition, among others [1,2]. However, many evolutions have been observed in recent years toward providing a wider range of products for consumers, with singular characteristics and properties. Minimally processed foods have shown characteristics similar to natural ones due to these developments in the aspects of engineering, nutrition, science, and technology. Accordingly, these foods have shown longer shelf life, nutritional functionality, different textures, and specific flavors, among other benefits [3,4,5].

In the same trend, the growing demand for biological products in agriculture encourages research on novel formulation techniques and especially the production of biological capsules [6]. This has occurred due to the stability of these bioproducts and higher reactivity of active ingredients, which minimizes volatility losses [7].

Based on this scenario, the encapsulation of bioactive substances (essential oils, plant extracts, fungal metabolites, etc.) in food and agricultural areas allows them to be protected against external factors and degradation. The encapsulation allows the biological integrity of the products and supports environmental conditions during storage, ensuring the viability of active ingredients for long periods [8]. For example, microbial agents are susceptible to abiotic and biotic factors that reduce the effectiveness of these living organisms and their metabolites when exposed to ultraviolet radiation and adverse temperatures, resulting in the loss of toxin integrity and spore viability [9]. In this sense, encapsulation is an alternative to these adversities. In the agricultural area, it allows for reducing losses by volatility, having better biological integrity, increasing efficiency, improving commercial viability, and increasing formulation stability, among others [10,11,12]. Therefore, the use of bioactive compounds as agents in the production of bioinsecticides, bionfungicides, bioherbicides, and biofertilizers is a promising strategy.

For encapsulation, many polymers are used as wall materials to protect the core, generally formed by bioactive compounds. Chitosan, gums (gum Arabic, Xanthan gum, gum acacia, and Shellac, for instance), maltodextrin, pectin, starch, whey protein, sodium alginate, cellulose and carboxymethylcellulose, zein, pullulan, galactomannan, and sodium caseinate, among others, are used for this purpose [13,14,15,16,17]. The polymers favor the retention of desired compounds in the systems formed during the processes and help to prolong the release of bioactive compounds for longer times or under specific conditions, such as certain pH ranges, for example [18,19].

Therefore, this review presents polymer matrices for the incorporation of bioactive compounds and discusses the characteristics of encapsulation systems. The data cited in the text were obtained from the scientific literature through the main scientific databases. The search was mainly focused on the five past years to have a recent scenario of findings and evolutions. Some figures were created by the authors to express the authors’ viewpoint and tables were compiled based on other referenced works.

## 2. Polymers Used for Encapsulation of Bioactive Compounds

Encapsulation is described as a process where a core material (a liquid, solid or gaseous compound) is packaged in a wall material to create capsules that are effective against chemical and environmental interactions [20,21]. Encapsulation is an alternative for problems of physical or chemical instability of compounds. It can inhibit volatilization and protect the encapsulated material against unfavorable environmental conditions, reducing the sensitivity to the degradation of plant materials and their bioactive compounds [22]. In such case, food biopolymers are used in the encapsulation process [23]. Biopolymers are classified into three classes: synthetic polymers derived from petroleum, synthetic polymers derived from renewable resources, and naturally produced renewable polymers, [24]. Many polymers (synthetic and natural) are used as a coating material for encapsulating bioactive compounds, such as polyethylene, polyethylene glycol, poly(vinylpyrrolidone), polyvinyl alcohol, polyacrylic acid, polylactic acid, polyhydroxyalkanoates, β-glucans, dextran, starch, alginate, cellulose, chitin, chitosan, pectin, collagen, gums, zein, hyaluronic acid, and gelatin, among others. Table 1 presents some examples of polymers and encapsulated bioactive compounds.

### 2.1. Chitosan

The deacetylation of the chitin polymer allows for producing chitosan, in which the acetyl groups of the chitin chain are removed to compose amino groups. The degree of deacetylation is directly interconnected with the performance of chitosan in its different uses [25]. Chitosan is a natural and non-toxic polysaccharide widely used due to its characteristics of biocompatibility, chemical resistance, and biodegradability. Moreover, it has the property of forming films without dependence on additives [26]. The high availability of chitosan is an important factor, which is the second most numerous natural polysaccharide biopolymer found in nature [27].

Chitosan still has antimicrobial and antioxidant activity [43]. The antimicrobial activity of chitosan depends on some parameters that affect its properties, such as pH, molecular weight, type of microorganism, source of chitosan, concentration, degree of deacetylation, complexes with certain materials, food components, and chitosan derivatives [25]. Chitosan is an excellent coating material used in the encapsulation of multiple bioactive compounds due to its characteristics and properties, having applications in food, pharmaceutical, agricultural, biomedical, environmental, and industrial segments, among others. This polymer is used in the encapsulation of essential oils, lipids, different food ingredients, vitamins, medicines, vaccines, hemoglobin, and microbial metabolites, among others [44]. In agriculture, chitosan and its encapsulated compounds have been widely used as an ecological alternative in agricultural production, such as biofertilizers, biopesticides, seed treatment agents, soil conditioners, and growth promoters [45]. Chitosan has been used as a co-encapsulating material for curcumin and resveratrol [46] and in the development of nanocomposite films to inhibit the growth of fungi, such as *Penicillium chrysogenum*, *Aspergillus flavus*, *Aspergillus niger*, and *Aspergillus parasiticus*, resulting in the control of these pathogens [27].

**Table 1 polymers-14-04194-t001:** Polymers used for encapsulation of bioactive compounds and the main objectives of the studies.

Polymer	Material	Encapsulated Bioactive Compound	Objective	References
Alginate	Ca alginate hydrogel granules	Reishi medicinal mushroom extract; Probiotic *Lactobacillus acidophilus*	Mask the bitter taste of extract and protect bioactive substances in oral administration to prolong cell viability under simulated gastrointestinal conditions and to protect the bioactive ingredients of Reishi mushroom along the storage	[16]
Alginate	Macrospheres	*Pseudomonas* sp. DN18	Effective protection against diseases caused by *Eclerotium rolfsii*	[28]
Alginate	Hydrogel	Jujube extract (*Ziziphu*s spp.)	Effect of encapsulation on antioxidant activity and protection of bioactive compounds	[29]
Gum Arabic	Adhesive membrane	Cinnamon extract	Present an active food packaging material with more control over its pungent smell and quick release	[13]
Gum Arabic	Nanocapsule	Savory essential oil	Alternative control method to the pre-emergence herbicide Metribuzin (Sencor^®^)	[30]
Gum Arabic + Chitosan	Nanocapsule	Saffron (*Crocus sativus* L.)	Increase bioavailability and protection of bioactive compounds through nanoencapsulation	[31]
Cellulose	Powder particles	Vitamin A	Evaluate the emulsifying properties of cellulose particles and the ability to encapsulate with vitamin A	[17]
Cellulose	Edible films	Probiotic bacteria (*Lactobacillus rhamnosus* GG)	Search for new applications of coatings and films based on edible cellulose as carriers of various probiotics	[32]
Celulose	Cryogels	Tebuconazole fungicide	Controlled release of Tebuconazole fungicide	[33]
Chitosan + Cellulose	Nanoformulations	Citronella essential oil (*Cymbopogon nardus* (L.))	Control of *Spodoptera littoralis*	[34]
Chitosan	Nanoparticles	Peppermint oil (*Mentha piperita* (L.))	A nanoinsecticide to control *Tribolium castaneum* (Herbst) and *Sitophilus oryzae* (L.)	[35]
Chitosan	Nanoliposomes	Caffeine	Retention and release of caffeine in the digestive system	[36]
Pectin	Hydrogels	Lactase	Lactase encapsulated in pectin-based hydrogels	[37]
Pectin	Film	Beetroot extract encapsulated in pectin from watermelon peel	Monitor the freshness of packaged chilled beef by developing a pH indicator film	[15]
Pectin	Edible coating	Carvacrol/2-hydroxypropyl-β-cyclodextrin (CAR/HPβCD-IC)	Fungal inhibition against *Botrytis cinerea* and *Alternaria alternata*	[38]
Shellac	Gels	Riboflavin *Lactobacillus acidophilus* amylase	Form shellac-based gels and oat protein at neutral pH as a carrier to entrap and deliver active substances	[19]
Shellac + Chitin	Composite microspheres	Yeast alcohol dehydrogenase (YADH)	Enzymatic immobilization by adsorption	[39]
Shellac + Zein	Composite capsules	Curcumin	Controlled release of curcumin	[40]
Xanthan gum + sodium alginate	Gels	Debranched pea starch	Improve the performance of debranched pea starch gels	[41]
Xanthan gum	Edible films	Xanthan Gum solutions with glycerol acid	Increase lotus root storage stability	[42]

### 2.2. Cellulose

Cellulose is a natural polysaccharide with properties of high interest, e.g., biodegradability and biocompatibility [17]. In terms of chemical composition, cellulose is a homopolysaccharide, which is formed by many monomers of glucose [47]. Cellulose is found in plants, microorganisms, such as fungi and bacteria, and algae [48]. It is the most abundant naturally occurring polysaccharide [27]. Methylcellulose, carboxymethylcellulose, hydroxypropylmethylcellulose, and hydroxypropylcellulose are synthetic derivatives of cellulose, which are also used for the encapsulation and manufacture of edible food films. These films are usually used to protect fruits and vegetables through the use of barrier layers [26].

In addition to the synthetic derivatives of cellulose, generally carried out by etherification or esterification of hydroxyl groups, it is possible to produce cellulose derivatives from agricultural and agro-vegetable residues. New biopolymers that have unique characteristics and properties compatible with synthetic polymers can be used for their replacement with non-degradable polymers, combining their applications with environmental preservation [49]. Cellulose can be used as nanocomposites. Especially, the nanocrystals can be used as reinforcing agents in the preparation of composite materials due to their characteristics such as high reactivity, high strength and modulus, rigidity, renewability, low density, biodegradability, and no toxicity. Cellulose nanocrystals can compose films, food packaging, fibers, hydrogels, and drugs. Also, it can be used as a transport vehicle for agrochemicals and in many other segments of biotechnology [50].

### 2.3. Starch

Starch is a natural polymer, which is the main polysaccharide reserve material existing in photosynthetic tissues and other parts of plants such as fruits, seeds, tubers, and roots. Therefore, depending on its botanical origin, starch can have a different molecular structure, as well as distinct characteristics and properties, in addition to exhibiting diversity in the shape, size and composition of its granules [51].

Starch is characterized as a polymeric compound consisting of glucose monomer, while it is present in abundance in foods, such as grains, corn, potato, and rice [26]. This polymer has 20–25% amylose and 75–80% amylopectin, presenting hydrophilic granules. It is a material with multiple applications due to its biodegradability, availability, and abundance, in addition to presenting competitive cost compared to other polymers [49].

### 2.4. Alginate

Alginate is a natural hydrophilic polymer isolated from brown algae and is used in food applications due to its film-forming, gel-making, thickening, and stabilization properties [26]. Alginate exhibits antimicrobial properties, moisture absorption, gelation, biocompatibility, and application in various segments (pharmaceutical, cosmetic, food, and biomedical industries) as a consequence of its multiple materials such as films, hydrogels, microspheres, fibers, and microcapsules, among others [52].

Alginate is also widely used with other polymers to improve or obtain new properties through the integration of materials, such as the production of edible films from the composition of alginate and fish gelatin, resulting in the improvement of the mechanical characteristics of the film and increasing antioxidant capacity [53]. The manufacture of bilayer films based on sodium alginate and tea tree essential oil incorporated in TiO_2_ nanoparticles is reported, aiming at improving postharvest quality and reducing anthracnose in banana fruits [54]. The use of alginate for the encapsulation of vegetable oils is quite widespread, such as the use of alginate-based microparticles structured with other biopolymers such as pectin to improve the encapsulation efficiency of essential oils from olive leaves [55].

### 2.5. Shellac

Shellac is a natural polymer refined from lac resin that is excreted from insects, mainly from the species *Kerria* spp. As it is a natural resin, shellac exhibits non-toxic and biodegradable properties. It is characterized as a semi-crystalline polymer but with less regular alignment, less dense, and brittle, presenting itself as an alternative to synthetic polymers. Moreover, it has potential in the area of green technologies [56].

Shellac is composed mainly of oxyacid polyesters, being an amphiphilic biomacromolecule. This biopolymer has been widely used in several segments due to its characteristic of amphiphilicity and pH responsiveness. Examples of applications are food coating, biodegradable films, manufacturing of food waxes, preparation of food delivery systems (coated carriers, microcapsules, nanoparticles, and microparticles), and as an oil gelling agent [57].

Many studies report the use of shellac with other materials to improve the efficiency of the delivery of bioactive compounds through a synergistic effect between the substances. One example is the combination of shellac and tannic acid to prolong the shelf life of post-harvested mango and improve its quality [58]. Another example is the interaction of oat protein granules and shellac to the development of gels at near neutral pH to protect and release sensitive bioactive compounds [19]. The preparation of microspheres composed of shellac and chitin to obtain high-efficiency enzymatic immobilization with great potential in biotechnology is also highlighted [39]. Furthermore, the development of films composed of shellac and gelatin to extend the shelf life of banana fruits and improve post-harvest quality has been studied, thus presenting satisfactory results [59]. The use of zein and shellac composite particles to improve curcumin encapsulation efficiency can be cited [40], among others.

### 2.6. Pectin

Pectin is a natural heteropolysaccharide found in the meristematic tissue and the parenchyma. The amount of pectin in the cell wall and its quality change according to the plant source [60]. Pectin is composed of poly α−1–4 galacturonic acid residues with degrees of methylation of carboxylic acid residues. It is also composed of sugars, such as arabinose, galactose, and rhamnose [61].

The degree of esterification is a factor that influences the processing conditions and pectin characteristics. It is the percentage of the number of esterified carboxyl groups in relation to the degree of methylation. Pectin is separated into high methoxyl pectin (degree >50%) and low methoxyl pectin (degree < 50%) [62].

Pectin is a food safety active ingredient and presents several benefits, such as gelling agent, emulsion stabilizer, and binding capacity agent. It is not digested by intestinal enzymes, while it is mucoadhesive [23,61,62,63]. This biopolymer has anti-inflammatory, non-toxic, biodegradability, and biocompatibility properties. It can be found as a hydrogel, fiber, composites, and injectable cell vehicle [52].

### 2.7. Gum Arabic

Gum Arabic is a natural source of mineral salts (potassium, magnesium, calcium), fiber, and carbohydrates (arabinose and galactose). It is a heteropolysaccharide often used as a coating material due to its characteristics, such as retention of volatiles, emulsification, low cost, negatively charged properties, low viscosity, high solubility, easy use, inhibition of oxidation reactions, and colorless characteristic of solutions [64,65]. Gum Arabic is mainly used in the pharmaceutical area to concentrate syrups. In the food industry, it is used as a stabilizer, while in the agricultural industry it is mainly used to produce microcapsules containing bacteria [65,66].

### 2.8. Xanthan Gum

Xanthan gum is a polysaccharide biopolymer consisting of glucose, mannose, and diglucuronic acid, which is produced using some inexpensive nutrients such as sucrose, sugarcane molasses, and whey [65]. Xanthan gum is a novel generation of extracellular metabolites produced by bacteria such as *Xanthomonas campestris* [67].

This gum is easily dissolved in cold and hot water and produces a concentrated solution, even in a small amount [65]. Xanthan gum is used in many segments such as chemical, petroleum, cosmetic, food, and agricultural products [68]. This gum is also used in some beverages, candies, and frozen and canned foods [67]. Factors, such as temperature, pH, carbon sources, high pressure, polymer concentration, and viscosity, in the presence of galactomannan influence the production of this gum [69].

### 2.9. Dextran

Dextran is a linear polysaccharide biopolymer that contains hydroxyl groups used for the covalent bonding of some organic groups, such as hydrophobic compounds [70]. The modified dextran becomes insoluble or soluble in water and can form self-organized nanoscale particles by increasing the degree of substitution [71]. This biopolymer can be used for the production of nanocarriers to encapsulate active materials with different hydrophobicities. Despite its great potential, the nanoencapsulation of food bioactive compounds using modified dextrans polymers should be better studied [72].

### 2.10. Milk Proteins

Protein-based polymers, in addition to presenting mechanical properties of films equal or superior to other polymeric materials, have excellent gas barrier properties, which are widely used for food packaging [73]. In this context, milk proteins are used, which can be separated into caseins and whey proteins [74]. Casein is the main and high-quality milk protein [75], found naturally in milk in micellar supramolecule structures [76]. It constitutes approximately 80% of the total milk protein, while whey proteins represent approximately 20% of the total milk protein, and are normally obtained as a by-product (whey) of cheese production [77].

Milk proteins demonstrate several favorable characteristics for biotechnological, food, and agricultural applications due to their multiple functional and structural properties, benefiting the development of macro, micro or nano structures. Among its various functional properties, the emulsifying and foaming properties are the most important [74]. Milk proteins are used as nutraceutical vehicles, acting in the stabilization of bioactive compounds in foods, in the delivery of nutrients, in the improvement of food quality (sensory, color, flavor, texture), and the improvement of food safety. These proteins are found in different structural forms and levels, such as nanofilms, hydrogels, microcapsules, nanocomposites, nanocoatings, and microspheres, among others [78].

In the agricultural area, the encapsulation of an insecticide, *Cydia pomonella* granulovirus used in organic arboriculture, was performed with sodium caseinate, a hydrophilic milk protein. Lipophilic polyglycerol polyricinoleate and sodium caseinate concentrations larger than 1 wt% were very interesting for applications because the encapsulation remained at a high level for at least 300 days [79]. In addition, the encapsulation of neem seed oil extract within polymeric shells formed by whey protein isolate/maltodextrin was performed using spray drying. Neem seed oil contains azadirachtin as its main bioactive component, a compound having efficient insect repellent and insecticidal properties. As a result of the study, encapsulation efficiencies for neem seed oil were demonstrated higher in smaller microcapsules, with efficiency values of 60–92%. The association of botanical insecticides within biopolymer cores such as milk proteins presents a remarkable potential for increasing agricultural production levels and reducing impacts on human health and the environment [80].

## 3. Encapsulation of Bioactive Compounds

Bioactive compounds can be found in vegetables, fruits, cereals, legumes, roots, rhizomes, and other plant sources. In this sense, to improve their stability and applicability in both food and agriculture, encapsulation is a feasible technique. Microencapsulation and nanoencapsulation are the two most used technologies to encapsulate bioactive compounds [81].

For the encapsulation process of bioactive compounds, the selection of the appropriate encapsulating material and the encapsulation technique should be considered. Some reported techniques are emulsification, anti-solvent precipitation, electrospinning, and coacervation, among others. Coating materials influence emulsion properties, such as droplet size, viscosity, stability, and powder characteristics. Nanoencapsulation presents a particle size of less than 1 micron, microencapsulation presents a particle size between 1 to 1000 microns, and microencapsulation produces a particle size higher than 1000 microns [82].

Microencapsulation protects core bioactive compounds in a heterogeneous or homogeneous matrix. The technique and the type of coating materials play a significant role in the feed and powder properties [83]. Droplets of liquid or small particles are coated by a polymer to produce particles, such as microspheres or microcapsules. They are used in the industry as pharmaceuticals, chemical, and food agents. Microspheres are dense matrix systems, while microcapsules are hollow internally possessing a reservoir system. The microparticles can present many morphologies, which depend on the properties of the core with bioactive compounds, coating material, and microencapsulation technique [84]. The main types of encapsulation methods, physical, chemical, and physical-chemical, are presented in Figure 1.

Emulsion techniques are rather used in the food industry. Recently, Pickering emulsions have gained importance because they have higher stability to coalescence [85]. The Pickering emulsion technique for preparing microspheres, microcapsules, and foams is broadly studied because it is easy to operate and economically feasible [86]. However, there are other emulsion techniques (Figure 2).

Encapsulation allows active ingredients such as polyphenols, carotenoids, pigments, fatty acids, phytosterols, probiotics, vitamins, minerals, and bioactive peptides to be trapped within a matrix of different carriers [87]. Generally, the stability of these bioactive compounds is low, and encapsulation generates a powder with higher stability against variations in temperature, light, pH, or oxygen, increasing the release rate of these active ingredients. 

### 3.1. Bioactive Peptides

Bioactive peptides are found in an inactive form in the structure of the original proteins and are activated after the cleavage of the proteins [88]. In most cases, the peptide chain comprises the amino acids arginine, proline and lysine together with hydrophobic residues. They can be classified as exogenous endogenous substances. Endogenous peptides are produced in different cell types, like neural, immune, or glands throughout the body. Exogenous peptides came from a variety of sources, such as food, supplements, and drugs [89].

Peptide fractions and protein hydrolysates can be used as nutraceuticals, functional foods, or ingredients in food products [90]. Some factors can affect bioactive peptides, limiting their application, e.g., pH, enzymatic degradation, low water solubility, interaction with the food matrix, hygroscopicity, and possible bitter taste. In this sense, encapsulation of peptides with some coating polymers can improve these aspects [91]. Some studies have presented that encapsulation of peptides increases their stability, bioactivity, solubility and sensory characteristics, and reduces hygroscopicity [82]. In addition, the various colloidal systems used to encapsulate peptides have presented good encapsulation efficiency and stability [92].

These bioactivities depend on the amino acid and the peptide sequence, which comprises 2 to 20 amino acids [93]. Bioactive peptides can be obtained by hydrolysis of protein of animal or vegetable origin, by chemical or enzymatic hydrolysis. Nowadays, peptides are mainly applied in the food industry to obtain functional and nutraceutical foods. Additionally, bioactive peptides have functional properties and can be used as additives, such as solubility, emulsification, foaming, and water/oil binding capacity.

Encapsulation techniques are a promising alternative to increase stability and control the release of peptides for food applications. Also, encapsulation of peptides in nanocarriers increases the sensory characteristics by masking their bitter taste, increasing the solubility, and decreasing the hygroscopicity. Some peptides can be used as natural food preservatives to prolong shelf life or incorporated into bioactive packaging in the form of coatings and edible films to improve the safety of some foods due to their antioxidant and antimicrobial properties. An overview of the bioactive peptides microencapsulation process and characterization, sources of the bioactive, coating materials, encapsulation method, process conditions and results in terms of process parameters, encapsulation efficiency, particle size, and retention of their functional properties is shown in Table 2.

### 3.2. Fatty Acids from Vegetable Oils

Vegetable oils rich in unsaturated fatty acids are prone to oxidation, which produces free radicals called hydroperoxides [83]. The encapsulation system reduces their degradation and/or masks certain undesirable, preserving the bioactive compounds and improving their controlled release, solubility, and bioavailability. Moreover, micro- and nano-encapsulation of extracts and essential oils can enhance their functional properties in new-generation foods [94]. Applications for the encapsulated essential oils have been emphasized, including antimicrobials, pesticides, natural insecticides, repellents, food packaging and taste preservation, and lipid oxidation [95,96,97].

### 3.3. Vitamins

Vitamin E is a fat-soluble vitamin from plants with health benefits [98]. This vitamin is made up of eight isoforms, including α-, β-, γ-, and δ-tocotrienol, and α-, β-, γ-, and δ-tocopherol [99]. The most prevalent vitamin E homolog in nature is α- tocopherol [100]. Tocopherols are found in high-fat foods such as nuts (almonds, peanuts, hazelnuts), seeds (sunflower, cottonseed), cereals (rice, oat, wheat), and vegetable oils (palm oil, wheat germ oil). Moreover, it is found in vegetables and fruit, such as spinach, tomatoes, green beans, broccoli, turnip, pumpkin, cranberries, carrots, avocado, kiwifruit, and raspberries [98,99,101,102].

Vitamin E is a weakly water-soluble bioactive component, which restricts its absorption in the gastrointestinal system and its total bioavailability. There is an interest in developing vitamin E fortified foods and beverages. Therefore, encapsulation is necessary to increase its absorption [103]. Furthermore, α-tocopherol is very susceptible to oxidative reactions, mediated by oxygen, light, heat, and free radicals [103,104,105].

Spray-drying is a known encapsulation technique because it allows for a continuous operating procedure. For improving vitamin E oral bioavailability, the spray freeze-drying method has shown to be effective [103]. The food industry uses encapsulated -tocopherol as antioxidants to extend the shelf life of fat-based bakery products [104]. Various studies have suggested that a vitamin E concentration beyond the advised daily allowance can have positive benefits on human health and may prevent reproductive, nervous system, inflammation, and immune system problems [106].

Recent studies on the encapsulation of bioactive peptides, fatty acids from vegetable oils, and vitamins (tocopherols) are presented in Table 2.

**Table 2 polymers-14-04194-t002:** Recent studies on encapsulation of bioactive peptides, fatty acids from vegetable oils, and vitamins (tocopherols).

Bioactive Compounds	Sources	Encapsulation Material	Encapsulation Method	Process Conditions	Objective	Results	References
Bioactive peptides	Milk casein hydrolysate	Pullulan	Electrospinning (encapsulation of various bioactive compounds in the form of nanosized fibers)	Pullulan at 100, 120, and 140 g kg^−1^	Improve the stability, and low bioavailability, masking the bitter taste	Production of clean bead-free peptides-loaded pullulan nanofibres at 120 and 140 g kg^−1^ with an encapsulation efficiency of 72–86% and a mean diameter of 60–133 nm	[107]
Antioxidant peptides	Fish hydrolyzed collagen	Liposome	Freeze-dried	The highest encapsulation efficiency was found in SPC-CHO-0.5% HC (*p* < 0.05) (85%); liposome stabilized with glycerol presented the highest efficiency (75%)	Improve stability and bioactivities	Lyophilized SPC-CHO-0.5% HC presented higher stability than lyophilized SPC-GLY-0.25% HC during storage for 28 days at 25 °C	[108]
Antioxidant peptides	Fish protein hydrolysate	Not described	Spray dried	Enzymatic hydrolysis and chemical methods; spray dried at 180 °C inlet temperature to obtain powder; stored at −18 °C	Retention of antioxidant properties and microstructure	Visceral protein hydrolysate prepared with pepsin had better quality regarding antioxidant characteristics and papain in nutritional aspect	[109]
Antihypertensive peptide	Whey protein hydrolysate <3 kDa	Alginate-collagen, alginate-gum Arabic, and alginate–gelatin	Extrusion method	Sonication for 15 min	Released antihypertensive peptides during gastrointestinal digestion	The highest efficiency was obtained in capsules of alginate—gum Arabic (95%); the released peptides incremented their ACE activity (85%)	[110]
Antioxidant peptides	Flaxseed protein	Maltodextrin	Spray drying	Hydrolysate to maltodextrin (MD) to ratios (1:1, 1:2 and 1:3, *w*/*w*); alcalase enzyme was used to produce hydrolysates	Retention of antioxidant properties and microstructure	Samples powders obtained by 1:3 ratio presented the highest radical scavenging activity for and ABTS+ (86%) and DPPH (69%); analysis of chemical structure indicated that hydrolysates were coated and dispersed within maltodextrin	[91]
Antioxidant peptides	Milk Casein hydrolysate	Maltodextrin	Spray drying	Spraying was carried out by a pressure nozzle, compressed air flow rate of 0.54 m^3^ h^−1^, flow rate of 5 mL min^−1^, and inlet air temperature and outlet air temperature of 130 ± 1 °C and 70.0 ± 0.5 °C	Reduction of hygroscopicity and retention of antioxidant properties	Antioxidant activities were 90–99%, 77–92%, 77–93%, 95–99%, and 77–98% after the spray-drying process; hygroscopicity was reduced by microencapsulation (*p* < 0.05)	[111]
Antioxidant peptides	Pink peach meat protein hydrolysate	Gum Arabic and maltodextrin	Emulsions-spray drying	Inlet air at 160 °C, outlet at 80 °C, nozzle diameter of 0.5 mm, air at 0.4 MPa, and spray flow feed rate of 15–20 mL min^−1^	Retention of antioxidant properties; improvement of sensory properties	Antioxidant activity was improved; sensory properties were improved in a concentration of up to 3%	[112]
Bioactive peptides	Azocasein	Not applicable	Double emulsions water-in-oil-in-water	Enzymatic hydrolysis	Improved bioavailability	Encapsulation efficiency of casein peptides was 93%	[113]
Antioxidant peptides	Casein hydrolysate	Gum Arabic and maltodextrin	Freeze-dried	Enzymatic hydrolysis, coating material (10:0, 8:2, 6:4); ultrasonication at 40 kHz, 750 W, 12 mm diameter tip and with 50% pulse for 20 min	Improve the antioxidant and sensorial properties	Reduced bitterness if compared to the casein hydrolysate; maintenance of antioxidant activity (93%)	[114]
Polyunsaturated fatty acids	Tea oil	Maltodextrin/Xanthan gum/Lysozyme nanoparticles	Pickering emulsion	Tea oil plus composite solutions at oil-water volume ratio of 1:5; homogenization for 3 min at 18,000 rpm to obtain the tea oil Pickering emulsion	Reduce lipid oxidation on tea oil powder	Encapsulation efficiency of 66% when 50% maltodextrin and 4% Xanthan gum/Xanthan gum/lysozyme nanoparticles was used; the surface of tea oil powder presented a relatively smooth porous microstructure	[115]
Omega-3 fatty acid	Flaxseed oil	Maltodextrin	Coacervation	-	Maintaining stability	-	[116]
Omega-3 and omega-6 fatty acids	*Linum usitatissimum*	Gum Arabic/whey protein/modified starch/sodium caseinate	Drying (spray drying, freeze drying)/supercritical emulsification/emulsion/coacervation	-	Prevents the oxidation of fatty acids	Microcapsules prepared with spray- and freeze-drying ranged between 10–400 and 20–5000 µm	[117]
Unsaturated fatty acids	Drumstick oil (*Moringa oleifera*)	Maltodextrin/gum Arabic (25:75); (oil to wall ratio 1:4)	Spray drying	Inlet air at 180 °C, outlet at 85 °C, air pressure of 0.06 MPa and air flow rate of 73 m^3^ h^−1^	To evaluate the protection of the encapsulating compound in drumstick oil	The range of emulsion droplet mean diameters was 1.94 to 4.92 µm; encapsulation efficiency of 91.05% with lower water activity; good oxidative stability; peroxide value was 7.63 to 8.07 meq of peroxide/kg of oil after 30 days of storage at 45 °C; particles size was 22.56 ± 0.63 µm;showed larger smooth-surfaced particles, which may indicate that viscosity of the emulsions and emulsifying capacity are higher.	[83]
Omega-3 fatty acid	Chia seed oil	Soy protein microparticles	Supercritical CO_2_-assisted impregnation	16 MPa impregnation pressure with ethanol as cosolvent (0.1-ethanol:oil ratio, *w*/*w*), temperature 40 °C and 4 h of contact time	Protect bioactive compounds through microencapsulation and increase bioavailability	Encapsulation efficiency of 95% and a retention efficiency of 35%, showing excellent oxidative stability; microcapsules ranged between 1 and 10 μm, having a spherical form with occasional depressions but no pores or fissures; 95.69 ± 4.28% of the encapsulated oil is released upon exposure to gastrointestinal conditions and becomes available for absorption.	[118]
Omega-3 fatty acid	Chia, camelina and echium oilseeds and wet microalgal lipids	Sodium caseinate and lactose (oil to wall ratio 1:4 (*w*/*w*))	Spray drying	Inlet air at 170 °C, compressed air pressure of 0.5 MPa, air flow of 700 L min^−1^ and aspiration 70%	Produce microencapsulated lipid extracts from sources of omega-3	Particles size ranged between 1.5 and 30 μm and they presented a spherical shape and a smooth surface without cracks; the chia fatty acid ethyl esters microcapsules had the best microencapsulation efficiency of 76.9%, while the echium microcapsules had the highest payload of 142 mg/g.	[14]
Omega-3 fatty acid	Flaxseed oil	4% gum Arabic and 16% soy protein isolate	Spray drying	Inlet air at 150 °C, outlet at 80–85 °C and flow rate of 4 mL min^−1^	Evaluate the effect of flaxseed oil nano-encapsulation on stability, physical, color, rheological, textural, and organoleptic properties in egg-free cake	Nanoencapsulated flaxseed oil used as an egg replacer in cakes; had the greatest percentage of omega-3 fatty acids (30%); particle size less than 100 nm; encapsulation efficiency of 72%; moisture content of 4% and peroxide value of 1.1 meq/kg.	[119]
Polyunsaturated fatty acids	Walnut oil	Fructooligosaccharide/soybean protein isolate (20% *w*/*w*)	Freeze drying	Temperature − 46 °C, pressure 4.1 Pa for 48 h	Evaluate the fructooligosaccharide/soy isolate protein	Microcapsules ranged between 121.51 and 162.02 µm; fructooligosaccharide reduces particle size and increases viscosity; microencapsulated walnut oil had a peroxide value of 26.84 meq/kg after 8 days of storage, compared to the walnut oil which reached a peroxide value of 74.56 meq/kg.	[120]
Alpha-linolenic acid	Perilla oil; (*Perilla frutescens*)	γ-cyclodextrin	Inclusion complex	The formation of pseudorotaxane complexes that precipitate in aqueous media	Evaluate the thermal stability and bioavailability α-linolenic acid from perilla oil	The complexes may serve as an effective supply of α-linolenic acid to raise plasma omega-3 fatty acid levels	[121]
Cinnamaldehy	Cinnamon essential oil	Carboxymethyl cellulose and polyvinyl alcohol	Pickering emulsions by in situ hydrophobization	Use oleic acid as a hydrophobic compound	Increase the shelf life of the bread	No fungal growth at 25 °C for 15 days; controlled release of cinnamon essential oil; fungal inhibition against *P. digitatum* in films containing 1.5 and 3% CEO.	[122]
Flavonoid karanjin	*Pongamia pinnata* L. seed oil	Polyuria	Interfacial polymerization	400–500 rpm slow mixing	Evaluate the insecticidal activity of microencapsulated *P. pinnata* oil	High encapsulation efficiency of 87.41%; release kinetics was y = −0.0042 x + 6.4205; effective protection against *Aphis gossypii* (71.8%) and *Bemisia tabaci* (74.7%).	[123]
Monoterpene α-pinene	Juniper berry essential oil	Gum Arabic/maltodextrin (1:1); (oil to wall ratio 1:4, *w*/*w*)	Spray drying	Inlet air at 120 °C, outlet at 80 °C and 3.2 cm^3^ min^−1^ of feed flow rate	Evaluate properties of microcapsules	Particle size was 10.83 µm ± 1.86; encapsulation efficiency of 70.07% and a retention efficiency of 82.66%; powder has the following characteristics: 4.92% moisture, 10.18% hygroscopicity, 63.80% solubility, 72.83% porosity, and 3.23 min of dissolving time; depending on the kind of wall material, it took between 15 and 45 min for the oil to completely discharge; presents antimicrobial and antifungal activity; it can be used as a food preservative.	[124]
Eugenol	Clove essential oil	Chitosan nanoparticles (oil to wall ratio 0.5:1)	Emulsification (*o*/*w*) and ionic gelation	Homogenize at 13,000 rpm for 10 min in ice bath conditions; for ionic gelation of the chitosan, sodium tripolyphosphate was added and agitated for 40 min	Improved antioxidant and antimicrobial activity by nanoencapsulation of clove essential oil	Particle size of 295.8 ± 45.6 nm; high retention rate (73.4%); high in vitro antimicrobial activity against *Listeria monocytogenes*, *Staphylococcus aureus*, *Salmonella typhi* and *E. coli* (a 4.80 to 4.78 cm inhibitory halo).	[125]
Alpha-tocopherol	Wheat germ oil	1.5% Sodium alginate and 2% pectin	Air atomization	O/W emulsion dropped; 5% (*w*/*v*) calcium chloride solution agitated for 30 min	Increase antioxidant activity and thus shelf life and nutritional value of cookies	Encapsulation efficiency of 55.97%; maximum antioxidant activity of 41.1%; improved storage stability and shelf life of cookies; microencapsulated α-tocopherol can serve as an antioxidant to avoid autoxidation in fat-based bakery products.	[104]
Alpha-tocopherol	Palm oil	Maltodextrin and sodium caseinate; (Core to wall ratio 1)	Spray drying	1.5 mm nozzle diameter, 10 mL min^−1^ of feed flow rate, 55 kgf cm^2^ air pressure, 20,000–25,000 rpm atomization speed, inlet air at 110 °C and outlet at 90 °C	Demonstrate the encapsulating and protective capacity of the wall material for the microencapsulation of vitamin E	Encapsulation efficiency of 59.9 ± 0.017 to 71.5 ± 0.027%; particle size from 13 to 29 µm; moisture content from 4.5 to 4.98%, microcapsule considered soluble due to the short solubility time of 178 to 251 s.	[126]
Alpha-tocopherol	Palm fatty acid distillate	Galactomannan and gum acacia	Spray drying	Inlet air at 180–200 °C and outlet at 90 °C	Evaluate emulsion and oxidative stability	Encapsulation efficiency between 60.68 and 70.01%; microcapsules ranged between 16 µm and 11 µm; a yield between 53.15 and 64.09%; moisture content from 3.40 to 3.08%; microencapsulation improved oxidative stability and absorption of vitamin E.	[127]
Alpha-tocopherol	Vitamin E	Whey protein isolate; (Core to wall ratio 1:3)	(1) Spray drying; (2) Freeze-drying; (3) Spray freeze-drying	(1) Inlet temperature 100 °C, outlet temperature 80 °C and 4 mL min^−1^ feed flow rate; (2) Temperature − 25 °C for 2 h and 7.6 × 10^2^ Torr to 0.8 Torr vacuum; shelf temperature between-25 °C and 20 °C for 16–18 h, after that 25 °C for 2 h;	Evaluate the effect of the three techniques of vitamin E microencapsulation	Encapsulation efficiencies and particle size of 90% and 195.8 µm for spray dried microcapsules, 86% and 279 µm for freeze-dried microcapsules, 89% and 145.3 µm for spray freeze-dried microcapsules, respectively; the rats showed plasma vitamin E concentrations of 7.35 at 4 h, 7.69 at 4 h, 9.45 µg/mL at 3 h; area	[103]
				(3) The temperature was kept between −25 °C and −10 °C with a vacuum of 0.8 torr, and then brought to 10 °C with a vacuum of 0.3 torr		under the curve were 109.84, 104.38 and 124.46 µg/(mL × h); spray freeze-drying microencapsulation improved the oral bioavailability by 1.13 and 1.19-fold compared to other techniques.	
Vitamin E	Palm oil	Maltodextrin/Sodium caseinate (3:2:1)	Freeze drying	Temperature −41 °C and pressure 4 × 10^−4^ mbar	Effect on vitamin E encapsulation with selenomethionine	Encapsulated vitamin E with 5.6 mg selenomethionine improves solubility and bioavailability; particle size was 3.00 µm ± 0.55; release rates of vitamin E after 30 min in simulated gastric fluid solution and simulated intestinal fluid solution were 87% and 42%, respectively.	[105]
Alpha-tocopherol	Vitamin E	Polycaprolactone	Supercritical fluid extraction of emulsions	Pressure 8 MPa and temperature 40 °C; CO_2_ flow rate of 7.2 kg h^−1^ kg^−1^ emulsion and acetone as solvent	Demonstrate the feasibility of supercritical fluid technique in the nano-encapsulation of liquid lipophilic compounds	High encapsulation efficiency of 90%; particle size was from 8 and 276 nm; spherical, core-shell, and non-aggregated nanocapsules were formed, according to morphological analyses; higher storage stability between 6 and 12 months.	[128]
Alpha-tocopherol	Vitamin E	Polycaprolactone	Nanoprecipitation	At 30 °C in an ultrasonic bath, PCL was dissolved in acetone, lecithin, acetone-methanol mixture (60 to 40%, *v*/*v*) and α-tocopherol	Improve the carboxymethylcellulose film in the production of active packaging with α-tocopherol nanocapsules	High encapsulation efficiency of 88.43–99.66%; particle size ranged between 201.6 and 230.2 nm; alpha-tocopherol nanocapsules’ release behavior from CMC films might be best described by the Higuchi kinetic model; the maximum radical scavenging activity (68.85%) was found in films with 70% nanocapsules.	[129]
Alpha-tocoferol	Vitamin E	Acid hydrolysis-carboxymethyl starch (H-CMS) and xanthan gum (XG)	Spray-drying	Inlet air temperature 190 ± 5 °C and outlet air temperature 80 ± 5 °C	Improved bioavailability	Microcapsules produced with substitution grades of 0.44 and a ratio of 1:20 (H-CMS/XG) demonstrated higher specific delivery in the small intestine, releasing 38.32% and 61.68% of vitamin E into simulated gastric and intestinal fluids, respectively	[130]
Alpha-tocoferol	Vitamin E	Gelatin and gum Arabic	Complex coacervation	Adjust the pH with acetic acid (10% *v*/*v*) to 4–4.5 (isoelectric point of gelatin and gum arabic), at a speed of 1500 rpm, temperature 45 °C for 90 min	Optimize by response surface methodology the conditions for vitamin E microencapsulation	High encapsulation efficiency (93.42%), when the core material is 4 g and surfactant is 0.5% (%*w*/*v*); particle size was from 4 to 80 µm.	[18]
Alpha-tocopherol	Vitamin E	Nano-hydroxyapatite as a Pickering stabilizer	Pickering emulsions	Uses a mixer with continuous mode; O/W ratio of 20/80 (*v*/*v*)	Improved bioavailability, bioaccessibility and stability	Particle sizes were 7.53, 11.56 and 17.72 μm; improved bioaccessibility of vitamin E by 10.87 ± 1.04% for gelatin and 18.07 ± 2.90% for fortified milk	[131]

### 3.4. Polyphenols

Polyphenols include a heterogeneous group of bioactive compounds containing at least one phenolic ring, naturally found in fruits, rhizomes, cereals, coffee, and tea, among others [132]. Anti-inflammatory, antioxidant, and anticancer properties have been associated with them, and they also participate in the prevention of diabetes, cardiovascular diseases, and cancer [133,134]. During its encapsulation, biopolymers are used, which stand out for being biodegradable, biocompatible, and high nutritional value vehicles. Among the most widely used biopolymers are carriers based on proteins, polysaccharides, and lipids [135].

Proteins represent excellent carriers of polyphenols due to their functional properties of emulsification, amphiphilicity, gelation, and foam formation. They can form nanoemulsions, nanogels, nanoparticles, nanofibers, and nanofilms, producing hydrophilic and hydrophobic polyphenolic compounds [136]. Likewise, polysaccharides, due to their structure and physiological activity, represent very suitable nanocarriers for encapsulating and administering polyphenols. Lipids represent one of the most important vehicles for the protection and supply of fat-soluble polyphenols due to their high biodegradability and biocompatibility. Through the use of liposomes, nanoemulsions, and solid lipid nanoparticles, the bioavailability of fat-soluble polyphenols can be improved [132]. In general, for the encapsulation of plant polyphenols, different types of biobased nanocarriers can be used, such as nanomicelles, nanoemulsions, nanoparticles, nanogels, and liposomes, including soy protein, albumin, zein, cellulose, starch, and lipids, which efficiently manage and protect polyphenolic compounds, as well as improve their bioavailability [132,134].

### 3.5. Carotenoids

Carotenoids are isoprenoid compounds found in plants, bacteria, fungi, and algae, as well as in foods such as fruits, vegetables, and fish. There are more than 600 fat-soluble carotenoids that differ by structural changes in their polyene skeleton, subdividing them into carotenoids, xanthophylls, and lycopene [137]. Different biological actions are attributed to it, such as important antioxidant activity in the prevention of age-related macular and cardiovascular diseases, as well as participating in the strengthening of the immune system, proper functioning of the reproductive system, regression of malignant lesions, and inhibition of mutagenesis [138]. They are sensitive to degradation by heat, light, and temperature. Likewise, the presence of oxidizing agents and oxygen could cause their decomposition [139], demonstrating the necessity to encapsulate them to stabilize and properly administer these bioactive ingredients.

Among the methods used for its preservation, microencapsulation is one of the most used. It allows stabilizing and reducing the volume of the product, thus reducing storage and transport costs, as well as favoring its administration. Among the oldest and most widely used carotenoid microencapsulation methods, the spray-drying allows for obtaining a product with the appropriate characteristics by varying the drying parameters such as feed flow rate and temperature, while different agents can also be used as carriers. Currently, among the most modern methods for the encapsulation of carotenoids, supercritical micronization is used, which allows for controlling the particle size and distribution of the particles, stability, and bioavailability of the product [138,140].

### 3.6. Pigments

Anthocyanins, carotenoids, betalains, and chlorophylls are natural pigments widely used in the food industry [138]. Among them, anthocyanins are water-soluble pigments that compose the vacuoles of several plant tissues, mainly in flowers and fruits, leaves, stems, and storage organs. The total content and composition vary substantially between different plant species [141,142]. However, they are susceptible to degradation during processing and storage due to pH, light, heat, oxygen, and interaction with other food components, decreasing their bioavailability and biological activity [143]. Microencapsulation using biopolymers based on proteins and polysaccharides of food origin is one of the most common methods to stabilize and protect anthocyanins from degradation. However, in a physiological environment, these systems could be unstable, due to high particle size and low efficiency of encapsulation of anthocyanins and zeta potential. Currently, for the stabilization of anthocyanins, nanosystems, such as nanoemulsion and nanoliposome, are being used. Unlike conventional emulsions, nanoemulsions provide higher stability against aggregation and gravitational separation due to their high surface area. Likewise, nanoliposomes made from conventional liposomes, in which the particle size has been reduced by ultrasound, high-pressure homogenization, maintained the stability, bioavailability, and controlled administration of anthocyanins [141].

Betalains are found in leaves, flowers, fruits, roots, stems, bracts, petioles, and seeds. They are synthesized from tyrosine and are divided into two subclasses: betaxanthins and betacyanins [144]. Antioxidant and anticancer properties have been attributed to them [145]. However, they are highly unstable and can be degraded according to variations in temperature, light, oxygen, and pH [146]. It has been reported that the encapsulation of betalains allows storage stability of up to 6 months. Moreover, the encapsulated betalains could be resistant to degradation and added to liquid or solid substances [147,148].

For the encapsulation of betalains, different matrices have been used, such as polysaccharides and proteins, or a combination of both, due to the high solubility of most polysaccharides and the high capacity of these compounds to protect betalains from oxidation [146]. Among the techniques used for the encapsulation of betalains, the most common are lyophilization, emulsions, ionic gelation and spray drying. The latter is the most used due to its cheaper cost and faster processing. However, it requires that the carrier be soluble in easily evaporable solvents at high temperatures, which could degrade some bioactive compounds. The lyophilization technique presents a higher loading efficiency and stabilization of betalains compared to spray drying. Likewise, it is reported that guar gum is suitable for encapsulating betalains with the lyophilization technique. However, it is not suitable for use with the spray-drying method [147,148].

### 3.7. Encapsulating Efficiency

Encapsulating efficiency is the ratio of a core material that is trapped inside an encapsulate as opposed to the initial core content that was introduced to an encapsulation process. It was determined using Equation (1) [149]:(1)$$\mathrm{Encapsulation}\;\mathrm{efficiency}\;\left(\%\right)=\frac{W_{a}}{W_{b}}\times 100\%$$
where $W_{a}$ represents the amount of integrated core material and $W_{b}$ represents the total amount of core material originally added during preparation.

Spectroscopic or chromatographic methods can be used to determine $W_{a}$ and $W_{b}$. Encapsulation method and particle size distribution can influence encapsulation efficiency [149]. The criteria that can affect encapsulation efficiency are wall and core specifications, core/wall ratio, emulsion properties (such as viscosity and droplet size) and drying parameters [150]. Encapsulation efficiency can be improved by selecting wall materials with different functional properties, and it is preferable to use various combinations of wall or shell materials [151,152]. Low efficiency might lead to low stability since capsules are not protected from adverse storage conditions. Currently, the primary focus of food component micro/nano-encapsulation is on enhancing encapsulation efficiency and prolonging product shelf-life [153,154].

### 3.8. Release Characteristics and Kinetics of Microcapsules of Bioactive Components

Bioactive components are released from encapsulants in three steps: surface release, diffusion via a swelling matrix, and matrix erosion [155,156]. More mechanisms were discussed, such as matrix degradation induced by enzymes and pH, fissure development, hydrostatic pressure, and geometric changes brought on by shear forces [155,157]. Surface release may be caused by inadequate entrapment inside the matrix or by a polar bioactive compound’s tendency to partition toward the hydrophilic surface of an emulsion [158].

Concentration-time profiles are frequently used to qualitatively evaluate release kinetics, with a focus on the quantity released after a certain length of time [113]. There have been several mathematical models developed, that describe the release of bioactive components. The following kinetic models were used: The zero order (Equation (2)), first order (Equation (3)), Hixson–Crowell (Equation (4)), and Korsmeyer–Peppas (Equation (5)) [155,159,160]:(2)$$Q_{t}=Q_{0\;}+k_{0}t$$
(3)$$Q_{t}=Q_{0\;}\cdot e^{-k_{1}t}$$
(4)$$Q_{0}^{1/3}-Q_{t}^{1/3}=Q_{0\;}+k_{H-C}t$$
(5)$$\frac{M_{t}}{M_{\infty }}=kt^{n};(M_{t}/M_{\infty \;}\leq 0.6)$$
where $Q_{t\;}$ is the quantity released after time t; $Q_{0\;}$ is the initial quantity which is usually zero; $M_{t\;}$ is the cumulative release at time t; $M_{\infty }$ is the cumulative release at infinite time; $k_{0}$, $k_{1}$, $k_{H-C}$, k represent the zero-order, first-order, Hixson–Crowell, and Korsmeyer–Peppas kinetic constants, respectively; n represents the diffusion exponent in the Korsmeyer–Peppas model. To characterize release from porous matrixes, the zero-order and first-order models are commonly utilized. In the Hixon-Crowell model, a linear graph of the cubic root of the unreleased fraction of capsule vs time demonstrates that the surface area of the composite changes during the release process [159]. According to R^2^ values, the Hixson-Crowell model was comparable to zero-order and first-order kinetics [155]. The Korsmeyer–Peppas model is typically utilized when the release mechanism is unknown or when the release process is regulated by more than one sort of release mechanism [159].

The releasing action is therefore known to be triggered by a range of internal (such as diffusion, degradation, and swelling) and external stimuli (such as changes in temperature, pH, light, ultrasound, ionic strength, and magnetic field). It is evident that numerous models have been established that may predict the release profile of active agents into the environment by taking into account the occurrence of different kinetic rates. Several researchers have attempted to forecast their experimental release data using various models such as the zero-order, first-order, Korsmeyer–Peppas model, and Hixson Crowell model. The Korsmeyer–Peppas model of bioactive compounds release is found to be used in the majority of investigations on the release of corrosion inhibitor from nanocontainers [161].

The encapsulation efficiency, release kinetics of bioactive compounds, stability of capsules, and matrix characteristics are all essential features that are affected by the technique and encapsulation materials [162]. The size of the encapsulating does not directly affect the efficiency of encapsulation, but it does affect the controlled release of bioactive compounds as well as the physical and chemical characteristics of encapsulants. This is due to the fact that smaller particle sizes result in bigger contact surfaces, resulting in more release than larger particle size encapsulates. Preferably, wall material should be insoluble, not react with bioactive components, good at producing films, and possess the appropriate protective qualities against a variety of external factors [162,163].

Finally, it has been found that most study include the encapsulation efficiency in their results, as it is one of the crucial parameters to assess the preservation of the core material (bioactive compound). However, there are few studies that cover the characteristics and release kinetics of bioactive compound microcapsules in food and agricultural applications, because kinetic models are more used in the realization of a simulation, evaluating the diffusion process that occurs through the nanocontainer, and making some assumptions appropriate for a specific application.

## 4. Novel Technologies for Encapsulation of Bioactive Compounds

The population is demanding healthy foods with higher added value, such as functional foods or foods that contain special nutrients. The food industry faces technological changes to meet the needs of consumers. For these reasons, the technology must be appropriate to process these types of foods. The micro- and nanoencapsulation of bioactive compounds, such as antioxidants, carotenoids, tocopherols, phenols, and bioactive peptides, among others, is mostly done by spraying. Currently, lyophilization is also used to microencapsulate bioactive compounds. Extrusion is another technique that has been used in this food and agricultural field.

In this sense, different techniques can be used to microencapsulate bioactive compounds. The selection of the technology depends on several factors, such as the types of bioactive compounds to be encapsulated, the sensitivity of the bioactive compound, the encapsulating agent, and the costs of the technique [164]. Each microencapsulation technique presents different results in the final product. The bioactive compound in the final product may have properties in terms of shape, structure, size, distribution, and bioavailability. Microencapsulation has some uses expressed in Table 3.

### 4.1. Supercritical Microencapsulation

Currently, supercritical microencapsulation is also known as “micronization”. Supercritical micronization uses mild temperatures, which avoids reducing the quality of bioactive compounds. CO_2_ is the most used solvent in this technique, followed by water [138]. CO_2_, under supercritical conditions, has become increasingly popular in the microencapsulation of bioactive compounds of interest to the food and agricultural industries [165,166,167].

**Table 3 polymers-14-04194-t003:** Technologies used for the microencapsulation of bioactive compounds.

Microencapsulation Technique	Method	Applications	References
Physical	Spray-drying	Phenolic acids, carotenoids	[122]
	Spray chilling	Pigments	
	Spray coating	Pigments	
	Supercritical microencapsulation—micronization	Carotenoids	
	Ionic gelation	Nutraceutical	
	Cocrystallization	Food ingredients	
	Freeze-drying	Carotenoids, Pigments	
	Fluidized bed coating	Carotenoids	
	Centrifugal extrusion	Food ingredients	
Chemical	Interfacial polymerization	Food	[138,166,168,169]
	Molecular inclusion	Nutraceutical	
	In situ polymerization	Nutraceutical	
Physical-chemical	Coacervation	Volatile flavor oils	[138,166,168,169]
	Complex coacervation	Lycopene	
	Emulsion-solvent evaporation	Food ingredients	
	Solidification emulsion	Food ingredients	
	Liposomes	Food ingredients, nutraceuticals	

### 4.2. Complex Coacervation

Complex coacervation is an encapsulation technique that consists of the interaction of two polyelectrolytes of opposite charges in an aqueous medium. The chemical complex of a protein or carbohydrate nature is formed around the bioactive compound to be encapsulated. This interaction is formed under specific conditions of ionic strength, temperature, pH, polymer concentration, the proportion of biopolymers, the molecular weight of biopolymers, and the degree of homogenization [168]. This technique allows for obtaining particles, where the core (bioactive compounds) is protected by a layer of an encapsulating agent of a protein or carbohydrate nature [169,170].

### 4.3. Spray Chilling

The application of vitamin C (ascorbic acid) has demonstrated beneficial effects in agriculture, such as reducing the effect of nickel and cadmium in barley [171,172], alleviating drought stress in sweet pepper [173] and Cucumber [174], and improving the qualities of apples [175]. It allows for mitigating the salinity stress and providing better growth performance in barley [176]. In this context, the encapsulation of ascorbic acid through spray chilling has been reported as an effective technique, showing retentions over 90% and controlled release of vitamin C in aqueous solutions [177,178]. These authors employed a combination of fully hydrogenated palm oil and palm oil to obtain flexible particles. The technique involves heating the fully hydrogenated palm oil to 80 °C to make it liquid and mixing it with the ascorbic acid to pass immediately through the spray dryer and be cooled to 5 °C.

### 4.4. Other Encapsulation Techniques

Some essential oils have the potential to be applied in agriculture as biopesticides. Rosemary oil has the potential to be used as an insecticide [179,180,181]. It has been encapsulated by spouted bed drying using a mixture of Tixosil and maltodextrin as drying carriers and Teflon beads for drying [182] and with microcrystalline cellulose cores [183]. Neem oil has shown insecticide activity [184,185], and it can be effectively nanoencapsulated to control pests [185,186,187], showing enhanced properties.

## 5. Applications of Encapsulated Products

Encapsulation materials consist of a variety of synthetic or natural polymers [188], the former still being the most used elsewhere. Otherwise, an increase in the use of natural polymers is seen, and non-hydrolyzed biopolymers have been widely used in agriculture due to their availability and low cost. Among them, hydrolyzed starches stand out as agents in the encapsulation of Bt pesticides (metabolites produced by *Bacillus thuringiensis*) because they protected environmental factors and improvement in the formulated product [7]. Moreover, encapsulation can be efficient for formulations of biofungicides, biopesticides, and/or biofertilizers in agricultural fields, becoming an economically viable technique for farmers [12]. The encapsulation can have several advantages in the agricultural sector [189], as shown in Figure 3.

Taking into account agricultural applications, [190] report *Trichoderma harzianum* as a biological control agent very sensitive to biotic and abiotic factors due to the presence of live spores. Consequently, encapsulation shows an improvement in activity against phytopathogens such as in the control of *Sclerotinia sclerotiorum* (white mold). In another study, spores of the *Trichoderma* species were encapsulated in biologically based lignin for the treatment of diseases of the vine trunk, demonstrated by in vitro tests that the spores remain at rest until germination is triggered by the fungus itself at the correct time [191]. Additionally, the fungus *Trichoderma* spp. was encapsulated in a sodium alginate matrix to guarantee good stability of the biopesticide formulations. The encapsulation strategy resulted in the survival of the fungi during the production and storage stages. After 14 months, the stored samples still had good conditions and viable cells [192].

In the case of microencapsulation by Pickering emulsion of *Metarhizium* conidia for the mortality of *Spodoptera littoralis* larvae, better cell distribution in leaves and control of the agricultural pest were observed [193]. Furthermore, in a study carried out, a formulation of *Bacillus thuringiensis* aizawai (BtA) encapsulated by Pickering water-in-oil emulsion was obtained. The efficiency of 92% in the mortality of *Spodoptera littoralis* larvae in the first instar was reported [194].

Similarly, it was shown that inoculating two strains of *Pseudomonas fluorescens* in potatoes using an alginate-gelatin capsule resulted in higher plant protection against harmful soil conditions and establishment in the rhizosphere [195]. Likewise, *Pseudomonas* spp. was encapsulated in sodium alginate and with salicylic acid containing zinc oxide nanoparticles, in which the efficiency in the antifungal activity against *Sclerotium rolfsii* was demonstrated [28]. Following this trend, a formulation of biopesticides with UV protection capacity in the encapsulation of fungal conidia in oil/water emulsion and stabilized by titanium dioxide (TiO_2_) was developed [196]. Satisfactory efficiency in their germination rate was achieved when exposed to natural UV light compared to unprotected conidia (not encapsulated).

Azadirachtin found in the neem tree (*Azadirachta indica*), which is a natural pesticide molecule, was encapsulated via nanoemulsification with whey protein isolate. This strategy demonstrated a positive effect on the mortality of *Spodoptera frugiperda*, a caterpillar that affects the soybean crop [10]. Moreover, the encapsulated essential oil extracted from orange peel effectively inhibited the growth of *Staphylococcus aureus* and *Escherichia coli* [197].

Regarding the encapsulation of biofertilizers, a study showed that formulations containing *Burkholderia cepacia* and *Pseudomonas fluorescens* encapsulated with phosphate alginate presented better growth conditions for wheat plants in semi-arid and salt-stressed areas [6]. The study demonstrated that biofertilizers consisting of *Pseudomonas fluorescens* and *Azosprillum brasilense* encapsulated with polymers of montmorillonite (clay mineral) and sodium alginate resulted in higher control of the formulation [11]. Furthermore, slow release of the active compounds was reached, thus contributing positively to the growth of the wheat crop with an increase in the biomass and aerial part of the plants.

Regarding the food industry, food encapsulation has been widely used over the years [5,115,198]. The encapsulation technique aims to protect and deliver the bioactive compounds to the target tissue of the human organism. This approach results in better stability and bioavailability of the bioactive compounds and increases their use and benefits for the human body [199]. Figure 4 shows some foods used in encapsulation technology. As presented in the scheme, the main interesting compounds for the food encapsulation technique are dyes and flavorings (candies), vitamins (vitamin A, D, E), antioxidants (citrus fruits and cereals), enzymes (lipase, invertase), bioactive peptide (milk), and polyunsaturated fatty acids (omega 3), and some mineral elements (iron, potassium), among others.

In addition to substances produced by microorganisms, substances found in herbal extracts and essential oils are also targeted to be encapsulated because they have antimicrobial and insecticidal properties, among others [21,200]. The control of the high volatility of oils and extracts is a challenge for biotechnology. Therefore, the encapsulation of essential oils and volatile extracts in the development of bioproducts for agricultural use is of great interest because the release of these compounds to the target agent is desirable to be slow and continuous [201].

Essential oil from savory leaves (*Satureja hortensis* L.), for example, was encapsulated with different natural polymers such as apple pectin, gum arabic, and gelatin. A higher encapsulation efficiency for all polymers was reported. Consequently, the efficiency of herbicidal activity in amaranth (*Amaranthus retroflexus* L.) and tomato (*Lycopersicon esculentum* Mill.) was increased [202]. Another active oil has been encapsulated and tested over time against bacteria oxidation: pepper oil. The encapsulation of such oil by gum arabic/maltodextrin resulted in an inhibitory effect against *Pseudomonas aeruginosa*, *Enterococcus faecali* and *Staphylococcus aureus*. The microencapsulation using these polymers contributed to a higher protection of pepper oil against the oxidation process during storage [203].

Curcumin extracted from turmeric was encapsulated by the Supercritical Anti-solvent (SAS) technique using various polymeric matrices to formulate a dye containing turmeric extracts. The encapsulating polymers tested were Eudragit^®^ L100 (Evonik-Germany), Pluronic^®^ F127 (BASF-Germany), and polyvinylpyrrolidone or mixtures of these materials. The dye formulation had a curcumin content of 4.45 µg mL^−1^ with a mean diameter of amorphous particles of approximately 5.7 µm. Also, the best dye solubility of 211 μg mL^−1^ of total curcuminoids was obtained at pH 4 [204]. Economic evaluations have also been developed in this area to provide information that can boost applications on larger scales and transfer the technology to the industrial scale. In this case, the technical-economic feasibility of the SAS technique for food applications was evaluated in terms of the precipitation of a vitamin complex containing riboflavin, δ-tocopherol, and β-carotene in zein microcapsules. The results showed that the average size of the microcapsules ranged from 8 to 18 µm, with spherical morphology, while the precipitation yield ranged from 410 to 820 g kg^−1^. The simulated manufacturing cost for the microcapsules ranged from USD 0.38 g^−1^ to USD 0.50 g^−1^, while these values may reduce with the possibility of reducing the mass flow of CO_2_ during precipitation [205].

Poly(ε-caprolactone) particles containing resveratrol were developed using the Gas Anti-solvent (GAS) technique, having heterogeneous characteristics. Encapsulation was positive because it did not change the chemical structures or the antioxidant activity of resveratrol. In addition, the microparticles maintained a constant release for 48 h and, in the thermal oxidative analysis, the difference between the samples and the control was 2.57 times smaller than the difference between pure resveratrol and the control sample [206]. Curcumin nanoparticles were also produced by encapsulation in Shellac by the GAS technique. After 2 months of storage at room temperature (25 °C), the aqueous nanodispersions of the particles with the encapsulated bioactive substance (Cur@Lac) showed long-term stability without agglomeration or sedimentation. Furthermore, the final retention rate of curcumin was over 85%. Colored drinks made from fruit syrup (pH = 4.0), fruit soda (pH = 6.0) and sparkling water (pH = 8.3) were prepared using Cur@Lac as a dye, which has a yellow-orange color [207].

## 6. Concluding Remarks and Future Trends

An overview of the latest research, technologies, and advances regarding the application of the encapsulation of bioactive compounds for food and agricultural applications is presented. The main forms of encapsulation are microencapsulation, followed by nanoencapsulation. A crescent interest exists in the development of micro or nanocapsules loaded with polyphenols, carotenoids, fatty acids, phytosterols, probiotics, vitamins, minerals, and bioactive peptides from natural sources, for the food industry. However, for applications in agriculture, the development of particles loaded with essential oils, lipids, phytotoxins, medicines, vaccines, hemoglobin, and microbial metabolites is the focus. The main purposes of the encapsulation strategy are to improve stability, solubility, bioavailability, sensorial properties, retention of bioactive properties and microstructure, reduction of hygroscopicity, and increase shelf life. The most widely used polymers for their high performance are gum Arabic, starch, and chitin, while the most commonly used technologies for encapsulation are emulsions-spray drying, emulsions-freeze drying, complex coacervation, followed by others, such as the emerging technology known as supercritical microencapsulation. More studies are needed on bioavailability for food applications and the effectiveness of bioactivity in agricultural applications, as well as scale-up studies at both pilot and industrial levels. However, the potential of encapsulation into polymeric matrices makes it a good strategy to protect bioactive compounds and broaden their use in both food and agriculture.

## Figures and Tables

**Figure 1 polymers-14-04194-f001:**
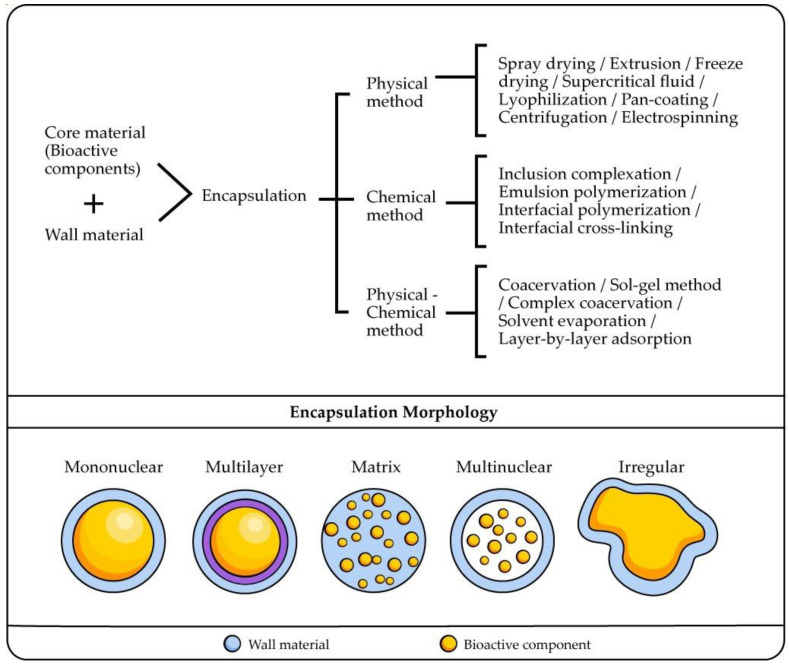
Encapsulation and morphology of microcapsule.

**Figure 2 polymers-14-04194-f002:**
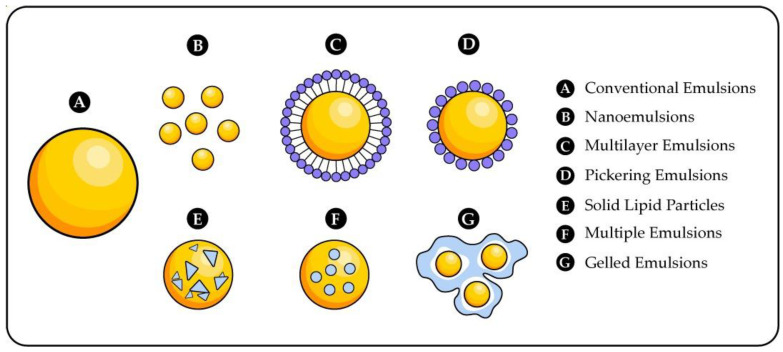
Schematic illustration of the emulsions.

**Figure 3 polymers-14-04194-f003:**
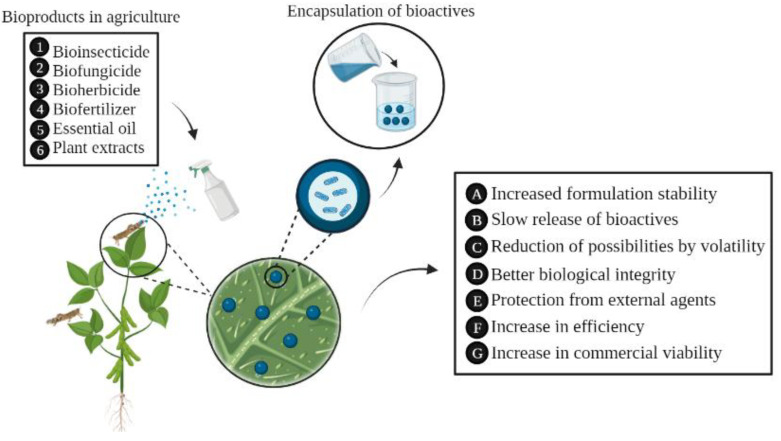
Advantages and applications of encapsulation in the agricultural sector.

**Figure 4 polymers-14-04194-f004:**
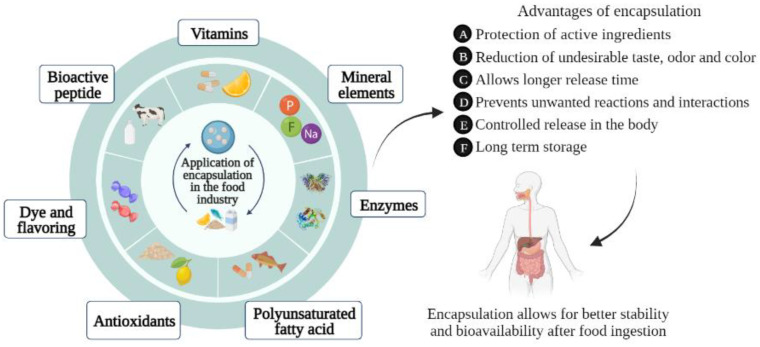
Advantages and applications of encapsulation in the food area.

## Data Availability

Not applicable.

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
