# Peer review of "Encapsulation of Bioactive Compounds for Food and Agricultural Applications"

_polymers, 2022, doi:10.3390/polym14194194_

Round 1

Reviewer 1 Report

The manuscript submitted to Polymers by Zabot et al. (manuscript number: polymers-1907547-peer-review-v1) mainly summarized the application of microencapsulation of bioactive ingredients in food and agriculture. The structural stability and efficacy of bioactive components are easily changed by light, heat, pH and other environmental factors and processing conditions. Microencapsulation is a useful way to protect the stability of bioactive components and improve their efficacy, and it is also a research hotspot in the field of food science. In this paper, the microencapsulation of bioactive components used in food and agriculture was reviewed in detail. However, there are still some shortcomings in the manuscript that need to be revised and improved by the authors.

1. In the Introduction section, the corresponding references need to be supplemented by the authors to support their views.

2. Milk protein polymers can also embed and microencapsulate bioactive components. It is suggested that the authors add research progress related to milk protein polymers in the manuscript.

3. It is also suggested that the authors should appropriately supplement the content related to the encapsulation efficiency, release characteristics and kinetics of microcapsules of bioactive components.

4. It is suggested that the authors should add an example diagram about the application of microencapsulated bioactive ingredients in food in the application section.

5. The application part should be as consistent as possible with the previous one: the text introduction after the subtitle.

6. In addition, the consistency between the title and the text should be confirmed in the manuscript, because all the materials introduced in the manuscript are bio-based, while the polymers written directly in the sub-title are too different.

Author Response

Dear Reviewer #1, thank you for your revision and suggestions for improving the manuscript. We have considered all the comments and changed the manuscript accordingly. The modifications are highlighted in red.

“1. In the Introduction section, the corresponding references need to be supplemented by the authors to support their views.”

Answer: Thank you for your suggestion. The corresponding references have been supplemented in the Introduction section.

“2. Milk protein polymers can also embed and microencapsulate bioactive components. It is suggested that the authors add research progress related to milk protein polymers in the manuscript.”

Answer: Thank you for your suggestion. Research progress related to milk protein polymers was added in section 2.10.

“3. It is also suggested that the authors should appropriately supplement the content related to the encapsulation efficiency, release characteristics and kinetics of microcapsules of bioactive components.”

Answer:   Thank you for your suggestion. The content on encapsulation efficacy, release characteristics and kinetics of microcapsules of bioactive compounds is supplemented in table 3 and points 3.7 and 3.8.

“4. It is suggested that the authors should add an example diagram about the application of microencapsulated bioactive ingredients in food in the application section.”

Answer: Thank you for your suggestion. The diagram was inserted in the application section.

“5. The application part should be as consistent as possible with the previous one: the text introduction after the subtitle.”

Answer: Thank you for your comments. The introduction and application section have been revised and modified to have an equilibrium between encapsulation for food and agricultural areas.

“6. In addition, the consistency between the title and the text should be confirmed in the manuscript, because all the materials introduced in the manuscript are bio-based, while the polymers written directly in the sub-title are too different.”

Answer: Thank you for your comments. The abstract, introduction and application section were revised and modified to have more consistency among these sections.

All authors have contributed significantly to the work and have read and approved the final version of the manuscript.

 Sincerely,

Dr. Luis OLIVERA

Universidad San Ignacio de Loyola

E-mail address: [email protected]

Reviewer 2 Report

The manuscript study "Encapsulation of Bioactive Compounds for Food and Agricultural Applications". The work is well-written and also the idea and information provided are interesting. However, there are some points that should be addressed. Below are some of the technical and non-technical points which should be addressed in order to move on:

Abstract:

It is better to specify the type of structures used for encapsulation in the abstract.

gum Arabic, starch and chitin

chitin of chitosan?

It is better to clarify whether the objectives of the study are about micro or nanocarriers?

It is suggested to mention the encapsulation methods that have been investigated in this study.

Why were these three (gum Arabic, starch and chitin) chosen? Considering that many other carriers are widely used for micro and nanoencapsulation.

The abstract should be clearly rewritten.

coating material

Is your study only about coating? If the mentioned polymers are meant to be used as carriers, it is better to mention them.

After reading the abstract, the reader should be aware of the goals, types of structures and methods of encapsulation described in this study.

1. Introduction

The first paragraph needs a reference.

Encapsulation methods should be mentioned in the introduction.

In the introduction, you mentioned only three types of polymers. But the text of the article contains a lot of polymers.

It is better to classify and present the encapsulation methods in a more separable way. Text coherence is important. Content should not be presented in a scattered manner.

The objectives of this study should be clearly reported in the abstract.

Author Response

Dear Reviewer #2, thank you for your revision and suggestions for improving the manuscript. We have considered all the comments and changed the manuscript accordingly. The modifications are highlighted in red.

“1. Abstract: It is better to specify the type of structures used for encapsulation in the abstract. Gum Arabic, starch and chitin, chitin of chitosan? It is better to clarify whether the objectives of the study are about micro or nanocarriers? It is suggested to mention the encapsulation methods that have been investigated in this study. Why were these three (gum Arabic, starch and chitin) chosen? Considering that many other carriers are widely used for micro and nanoencapsulation. The abstract should be clearly rewritten. Coating material. Is your study only about coating? If the mentioned polymers are meant to be used as carriers, it is better to mention them. After reading the abstract, the reader should be aware of the goals, types of structures and methods of encapsulation described in this study.”

Answer: Thank you for your questions and comments, questions and suggestions. The abstract was revised and completely rewritten to make clearer based on your appointments.

“2. Introduction: The first paragraph needs a reference. Encapsulation methods should be mentioned in the introduction. In the introduction, you mentioned only three types of polymers. But the text of the article contains a lot of polymers. It is better to classify and present the encapsulation methods in a more separable way. Text coherence is important. Content should not be presented in a scattered manner. The objectives of this study should be clearly reported in the abstract.”

Answer: Thank you for your comments and suggestions. The abstract and introduction were revised and modified accordingly to have more coherence. References were added in the Introduction section.

All authors have contributed significantly to the work and have read and approved the final version of the manuscript.

 Sincerely,

Dr. Luis OLIVERA

Universidad San Ignacio de Loyola

E-mail address: [email protected]

Round 2

Reviewer 1 Report

The author's reply and the improvement of the manuscript are satisfactory, and the revised manuscript has reached the level of being recommended for acceptance.

Reviewer 2 Report

accept